# Application of Hydrogels as Sustained-Release Drug Carriers in Bone Defect Repair

**DOI:** 10.3390/polym14224906

**Published:** 2022-11-14

**Authors:** Yujie Feng, Weiwei Guo, Lei Hu, Xuedong Yi, Fushan Tang

**Affiliations:** 1Department of Clinical Pharmacy, Key Laboratory of Basic Pharmacology of Guizhou Province and School of Pharmacy, Zunyi Medical University, Zunyi 563006, China; 2Key Laboratory of Basic Pharmacology of Ministry of Education and Joint International Research Laboratory of Ethnomedicine of Ministry of Education, Zunyi Medical University, Zunyi 563006, China; 3The Key Laboratory of Clinical Pharmacy of Zunyi City, Zunyi Medical University, Zunyi 563006, China; 4Department of Pharmacy, Affiliated Hospital of Zunyi Medical University, Zunyi 563000, China

**Keywords:** hydrogel, bone defect repair, bone tissue engineering, sustained-release drug carrier, self-assembling peptide

## Abstract

Large bone defects resulting from trauma, infection and tumors are usually difficult for the body’s repair mechanisms to heal spontaneously. Generally, various types of bones and orthopedic implants are adopted to enhance bone repair and regeneration in the clinic. Due to the limitations of traditional treatments, bone defect repair is still a compelling challenge for orthopedic surgeons. In recent years, bone tissue engineering has become a potential option for bone repair and regeneration. Amidst the various scaffolds for bone tissue engineering applications, hydrogels are considered a new type of non-toxic, non-irritating and biocompatible materials, which are widely used in the biomedicine field currently. Some studies have demonstrated that hydrogels can provide a three-dimensional network structure similar to a natural extracellular matrix for tissue regeneration and can be used to transport cells, biofactors, nutrients and drugs. Therefore, hydrogels may have the potential to be multifunctional sustained-release drug carriers in the treatment of bone defects. The recent applications of different types of hydrogels in bone defect repair were briefly reviewed in this paper.

## 1. Introduction

Bone is a primary part of the human body that provides mechanical support, protects internal organs and maintains the balance of calcium and phosphorus in the body [1,2]. Bone is a highly dynamic and vascularized tissue composed of bone substance, bone marrow and periosteum. Among them, the periosteum is the dense connective tissue rich in blood vessels and nerves covering the surface of bones. It can thus nourish bones and plays an important role in the treatment of nonunion and bone defects [3,4]. Bone defects caused by trauma, infection and tumors are common diseases in orthopedics [5,6]. Timely and appropriate treatments will help to reduce the incidence of nonunion and bone defects and improve the quality of the patient’s life. Bone has a powerful self-healing capacity [7,8]. Local bone defects shorter than critical-size defects (CSDs) can spontaneously regenerate and heal. However, in most cases, the bone defects are too large to regenerate themselves and have to rely on surgical intervention and bone grafting materials [2,9,10]. At present, there has been massive research on the clinical approaches to treat bone defects, mainly including bone transplantation, distraction osteogenesis (DO), Masquelet’s induced membrane technique (MIMT) and bone tissue engineering (BTE). Other than bones and bone substitutes, biofactors, together with small molecule drugs, also play an increasingly crucial role in bone defect repair. However, effective control of the release of biofactors and small molecule drugs remains one of the basic problems to be solved in bone defect repair so far. The appropriate dosage and release rate of biofactors and small molecule drugs are very important for bone tissue regeneration and local sustained-release drug delivery systems, which have been often used to control the release of the biofactors and small molecule drugs to delay their degradation [11,12,13,14,15]. Hydrogels have been a research hotspot of biomedicine owing to their good biocompatibility, controllable degradation rate and unique three-dimensional network structure and hydrophilicity, which enable them to simulate a natural extracellular matrix (ECM) and provide a suitable microenvironment for cells [16]. Therefore, hydrogels have considerable potential as cell scaffolds and drug delivery carriers in tissue regeneration [17,18].

This review aimed to illustrate the use of hydrogels as sustained-release drug carriers in bone defect repair. We briefly introduced the mechanism of bone defect repair and reviewed the common strategies currently used in clinical bone defect repair, with an emphasis on natural degradable hydrogel materials and synthetic degradable hydrogel materials recently used in bone defect repair. Prospects and challenges of the development of self-assembling peptides as drug delivery carriers for bone defect repair was reviewed last.

## 2. Bone Defect Repair

Bone is a highly dynamic and complex organ, and the balance between bone formation by osteoblasts and bone resorption by osteoclasts is well-coordinated under healthy physiological conditions [19,20]. Bone resorption is very important for maintaining mineral homeostasis, regulating function and bone defect healing, and is mainly affected by the number of osteoclasts [21]. However, this equilibrium is disturbed by many bone diseases such as trauma, infection and tumors. Bone defects refer to the loss of bone substance and are mainly manifested as the destruction of bone integrity, pain at the defect site and limited mobility. The repair of bone defects is a continuous process involving a series of biological events and a variety of osteogenic-associated cells (Figure 1). Often, bone defects are caused by various reasons, then inflammation develops at the site of the bone defect. This process produces a large number of cytokines, such as proinflammatory interleukins (IL-1, IL-6, IL-11 and IL-18) and tumor necrosis factor-α (TNF-α), that are involved in inflammation [22]. Various osteoblast-associated cells such as osteoclasts and mesenchymal stem cells (MSCs) are activated by these cytokines and are recruited to the bone defect sites. These cells proliferate at the site of the bone defect, at which point the stem cells further differentiate into osteoblasts to promote osteogenic differentiation and angiogenesis near the fracture site. Together with the osteocytes present in the bone, these cells promote bone repair by secreting various osteogenic-related matrices such as glycoprotein, osteocalcin and type I collagen [23,24].

## 3. Current Clinical Approaches in Bone Regeneration

The clinical treatments include bone transplantation, distraction osteogenesis (DO), Masquelet’s induced membrane technique (MIMT) and BTE. At present, the clinical approaches to large-scale bone defects mainly rely on bone transplantation, such as autografts, allografts and xenografts. With remarkable bone conduction, osteoinduction and osteogenesis properties, the autografts derived from the patients themselves have been widely used and are still regarded as the best choice for the treatment of serious bone defects [6]. However, due to the limited source of materials and a series of complications such as postoperative infection and pain, other methods are required to treat serious bone defects and must be considered [11,25,26,27,28]. Allografts and xenografts are often used as secondary options to overcome the problem of limited material sources. Unfortunately, both of these materials are antigenic, and their strong immune rejection and weak osteogenic capacity may eventually lead to bone graft failure while increasing the risk of infection in the recipient [29]. Some researchers thus do not recommend these materials for bone defect repair. DO technology [30] refers to the tension generated by the slow pulling of some biological tissues, which can stimulate tissue regeneration and active growth, and eventually, the newly formed bone tissue forms normal bone tissue after osteogenesis and mineralization [31]. The disadvantage of this technology is that continuous traction may cause pain at the surgical site of the patient, and a series of adverse reactions such as broken nails, infection and nonunion may occur [32]. MIMT is a relatively new, two-stage surgical procedure for the reconstruction of segmental bone defects [33]. In the first step, polymethylmethacrylate (PMMA) bone cement is used as an inducer to induce periosteum formation at the local defect site before bone transplantation. The induced membrane can wrap the bone in the later stage, induce angiogenesis, prevent the invasion of non-osteoblasts and promote the proliferation and migration of osteoblasts. In the second step, the PMMA spacer was removed 2 months later, and the cavity was filled with fresh autologous cancellous bone [34,35]. However, this technology has an obvious disadvantage: requiring two operations, which causes secondary trauma to patients and increases the infection rate [36,37].

Most of the current strategies have shown a certain performance in promoting bone defect repair and regeneration. However, each of them has more or fewer shortcomings. Therefore, it is necessary to develop a new method to replace or supplement the existing methods to overcome original limitations. Tissue engineering is the construction of vital cell–biomaterial complexes at cellular and molecular levels to replace defective or dysfunctional tissues and organs in terms of morphology, structure and function to form tissues and regenerate organs [38]. Bone tissue engineering is mainly composed of three elements: seed cells, scaffold materials and cytokines [39]. A perfect scaffold material for bone regeneration should have osteoconductivity, osteoinductivity and appropriate mechanical strength to act as a temporary scaffold to ensure the normal growth and reproduction of seed cells on the scaffold material. Various biomaterials have been identified and used for bone regeneration. The most frequently used are ceramics, synthetics and natural polymers and composites. One of the most commonly used biomaterials is the inorganic matrix of bone, even though it exhibits low tensile strength. It is formed from calcium phosphate. Hydroxyapatite (HA) is also widely used in bone injury repair because it is similar to the inorganic matrix of bone, but it takes a long time to reabsorb. β-tricalcium phosphate (β-TCP) is easier to produce and has a faster reabsorption time than HA, though it is more fragile [20,40]. Loading different biofactors, drugs and seed cells on the scaffold materials have achieved a major advancement in bone tissue engineering. Generally speaking, the large porosity makes it easier to transport nutrients and metabolic waste, which facilitates the growth and proliferation of osteoblasts. The gradual degradation of the scaffold material also allows for tissue formation at the site of the bone defect to achieve a permanent repair [16,41,42].

None of the materials used to treat bone defects have superior or even the same biological or mechanical properties compared with natural bone currently [43]. Therefore, it has always been a research hotspot to find ideal materials which can very well mimic natural bone tissue with good biocompatibility and long-term stable osteogenesis. Hydrogels may be one of suitable candidates in the related biomedical fields [44].

## 4. Research Progress of Hydrogel

Hydrogels are highly hydrated networks fabricated from a wide range of hydrophilic polymers, which can be classified as natural, synthetic or mixed according to the source of hydrogel materials. The hydrogel can retain a large amount of water by absorbing and swelling, similar to ECM. Because of their high-water content and strong biocompatibility, hydrogels usually do not result in irritation and hemolysis when they are introduced into human tissues or blood. In the past few decades, the preparation and application of hydrogels have shown far-reaching prospects, especially in the field of biomedicine.

Hydrogels have been extensively researched and applied in regenerative medicine [45,46,47,48], 3D cultures [49,50,51] and sustained-release drug delivery systems [48,52,53,54]. Skin regeneration is a great clinical challenge for wound healing. Hydrogels may become a promising skin regeneration material due to the following advantages: it can provide a moist, slightly acidic environment for the wound; it is not sticky and thus easy to remove; it seldom causes secondary harm to the patient; it can act as a barrier to prevent bacterial infection when directly contacting the human tissue [53,55,56,57]. Hydrogels are highly similar to ECM; most hydrogel materials can also be cross-linked under mild conditions to encapsulate living cells. Compared with 2D cultures, 3D hydrogel networks make cells interact with the environment in all directions, which simulates the growth environment of tissue cells well [58] and provides mechanical support for the cells [40]. Because of these features, hydrogels have been widely used as 3D ex vivo tissue models [58,59]. Oral administration and intravenous administration are the most common methods of clinical administration. However, some therapeutic drugs have first-pass effects and are easily degraded by enzymes, and oral administration has a slow onset. Although systemic intravenous administration has a fast onset, with the drugs directly flowing into the blood, the chance of adverse reactions is high [60]. Hydrogels also have great application in drug delivery, as they have good biocompatibility and also a high degree of physical and chemical regulation, which can be used to easily control the release of the molecule by changing the network size or their degradation rate. Through diffusion, swelling or other release mechanism based on certain environmental stimuli, sustained-release drug delivery can be achieved with hydrogels [61]. Because of the superior loading and release performance as a carrier for many kinds of drugs, hydrogels have been widely used in sustained-release drug delivery for bone defect repair (Figure 2). The next section will review in detail the application of different types of hydrogels as drug delivery vehicles in bone defect repair.

## 5. Application of Hydrogel in Bone Defect Repair

Many kinds of natural and synthetic hydrogel materials have been developed, including but not limited to collagen, chitosan, gelatin, alginate, hyaluronic acid (HA), fibrin, silk fibroin, poly (ethylene glycol) (PEG) and self-assembling peptide [16]. This section will review some biodegradable materials commonly used in bone defect repair, hoping to provide useful references and inspirations for future research in this field (Table 1 and Table 2).

### 5.1. Collagen

Collagen is the most abundant protein (by weight) in mammals, mainly distributed in connective tissue, tendons, ligaments and cornea. It also comprises the matrix of bones and teeth [86]. Collagen first forms collagen fibrils, which then further self-assemble into collagen fibers and finally form a hydrogel in the presence of a water-based solvent. These collagen fibers are randomly oriented in the absence of special treatment [87]. Types I, II and III are the most common and have considerable biodegradability, biocompatibility and the potential for the reconstruction of blood vessels [62,63,88]. Collagen I has been widely used for wound healing, drug delivery systems and tissue regeneration because it can repress inflammation and immune repulsion-derived side-effects [62,63,89,90]. Three-dimensional collagen-based scaffolds are also promising as drug/growth factor-activated matrices due to their highly porous properties, which allow cells to migrate rapidly to the central area, resulting in the effective uptake of target drug and growth factors encapsulated in the scaffolds. However, its relative degradation rate can cause a loss of mechanical strength, limiting its application in the regeneration of bones [20]. Previous research has demonstrated that the addition of graphene oxide (GO) to these hydrogels improves their mechanical properties by establishing chemical bonds with natural polymers [91,92]. Alendronate (Aln) can promote the osteogenic differentiation of MSCs and inhibit osteoclasts [74,75]. Zeng [74] loaded Aln into a type I collagen (Col) cross-linked with GO to fabricate (Col-GO-Aln) sponge for bone defect repair. Chen [76] designed a novel enzyme cross-linked recombinant human procollagen-like (HLC) sponge containing recombinant human BMP-2 (rhBMP-2) (HLC-BMP hydrogel), which has strong mechanical strength, controllable biodegradation and a unique vertical pore structure (Figure 3A). This makes it conducive to the migration and angiogenesis of mesenchymal stem cells (MSCs) to the HLC sponge (Figure 3C). In a rat cranial defect repair model, they then used this specially structured HLC sponge loaded with a low dose of rhBMP-2 to achieve a short burst release of BMP-2 followed by a steady slow process (Figure 3B), which can significantly induce bone regeneration in situ and improve bone repair percentage within 4 weeks (Figure 3D).

### 5.2. Chitosan (CHI)

Chitosan is a polysaccharide formed by deacetylating chitin in an alkaline medium, which is mainly derived from the shells of crustaceans such as crabs, shrimps and lobsters [64,93]. The positively charged protonable amino group of the d-glucosamine residues of chitosan can interact with the negatively charged sialic acid residues of the glycoprotein, which comprises the mucus. Because of its inherent antimicrobial properties and good biocompatibility, chitosan has been widely used in drug delivery, wound healing and tissue engineering.

Chitosan also has the disadvantage of poor mechanical strength, which can be partly improved through chemical bonding, cross-linking agents and photopolymerization. Chitosan is frequently associated with natural polymers of plant or animal origins. For example, phytochemical-grafted chitosan (PGC) is synthesized by grafting caffeic acid to the amino group of ethylene glycol chitosan (GC) [94]. Afterward, Lee [95] prepared a nano clay-organic hydrogel bone sealant (NoBS) which is formed by the self-assembly of PGC with nano silicate NCs. The hydrogel system has stronger mechanical properties, biocompatibility, injectability, antibacterial activity and osteoinductive activity. The NoBS system showed great osteoinductive capability by regulating the Wnt/β-catenin pathway, and it also promoted bone regeneration in critical-sized mouse calvarial defects and can be an ideal bone substitute for new treatment methods in bone tissue engineering.

While cells themselves are not drugs, it is also a feasible method to deliver stem cells with bone differentiation ability to the bone defect site for bone repair. Homing factors are usually needed to overcome the limitations of seed cells in the process of cell expansion and passage. Among various homing factors, stromal cell-derived factor-1α (SDF-1α) can induce host stem cells to migrate home to the injured site through the cell homing pathway and promote bone regeneration. Mi [79] prepared SDF-1α/CS/Carboxymethyl chitosan (CMCS) NPs incorporated CS/β-glycerol phosphate disodium salt (CS/GP) hydrogels, which displayed a crosslinked network with regular pores and a porous structure. SDF-1α/CS/CMCS NPs with a diameter of 100~200 nm was embedded into the pore diameter of CS/GP hydrogels, which was about 150 μm (Figure 4A). The hydrogel can control the release of SDF-1α in situ after an initial burst release of about 20% in the first 24 h and a cumulative release of 40% after 28 d incubation to solve the problems of degradation by enzymes and the short half-life of SDF-1α (Figure 4B). The Micro-CT of rat calvarial defects model revealed that the SDF-1α/CS/CMCS NPs embedded hydrogels group could significantly promote the new bone formation compared to that of the SDF-1α embedded hydrogels group and the control group (Figure 4C). This kind of composite hydrogel may provide a safer and more effective method for bone repair and regeneration.

### 5.3. Gelatin

Gelatin is obtained by the hydrolytic degradation of natural collagen and is the main component of connective tissue. Gelatin is soluble in warm water (>40 °C) but can form thermoreversible hydrogels when cooled, so it can be used for preparing temperature-sensitive hydrogels. Because of cell adhesion ability due to its rich RGD sequences [65] and many other advantages such as low price, good biocompatibility, degradability and weak immunogenicity, gelatin has been widely researched and applied in biomedicine, including contact lenses, artificial tendons, tissue engineering and drug delivery systems [96,97].

Gelatin alone also has insufficient mechanical strength as a bone defect repair material. In order to improve the mechanical properties of natural gelatin hydrogels, the following three methods can be used: ① Adding a crosslink agent. The addition of crosslinkers is one of the most common ways of enhancing the mechanical properties of gelatin hydrogels. Skopinska-Wisniewska, J [98] treated gelatin hydrogels with three cross-linking agents (EDC-NHS, Squaric acid and dialdehyde starch) to enhance the gelatin hydrogel properties. The results showed that EDC-NHS and dialdehyde starch increased the tensile strength, elongation at break and stiffness of the gelatin hydrogels and significantly improved the mechanical properties of the gelatin hydrogels compared to those without the cross-linking agents. ② Form hybrid gels. Currently, there are frequently reported hybrid gels such as bioactive glass chitosan/gelatin gels [99,100], bioceramics/gelatin gels [101,102,103] and polymers/gelatin gels [104]. For example, graphene oxide (GO) can not only promote osteogenic differentiation of mesenchymal stem cells [105] but also promote angiogenesis [106] in vitro with the activation of phospho-eNOS. The abundant amino groups on the surface of gelatin can reduce graphene to graphite oxide (GOG). Gelatin has an obvious reducibility due to the abundant amino side chain on its backbone of the molecular chain. Asparagine (Asn), Glutamine (Gln), Arginine (Arg) and Lysine (Lys) with –NH_2_ groups in the gelatin may interact with the –COOH in the surface of GO to constitute GOG, which showed better stability. Some studies have found that GOG can synergistically promote osteogenesis with Wnt/β-catenin agonist methyl vanillate (MV). The abundant amino groups on the surface of gelatin can reduce graphene to graphite oxide (GOG). Some studies have found that GOG can synergistically promote osteogenesis with Wnt/β-catenin agonist methyl vanillate (MV). Jiao [107] prepared a PH/GOG complex (PH/MV/GOG) loaded with MV. The results showed that the composite hydrogel system could mediate the two-way differentiation of BMSCs into bone formation and angiogenesis and release MV from the hydrogel. It also promotes bone healing by initiating the osteogenesis of BMSCs. ③ Covalent modification to the gelatin structure. Chen [77] constructed a range of gelatin hybrid hydrogels (Gel-POSS hybrid hydrogels) containing different concentrations of polyhedral oligomeric silsesquioxane (POSS) by esterification reactions. And a microporous structure similar to gelatin hydrogels was observed in all the hydrogels. SEM results show that smaller pore sizes were formed with increasing POSS concentration. This means that the addition of POSS provides more cross-linking points for the hydrogel, resulting in smaller pore sizes and enhanced mechanical properties (Figure 5A,B). In addition, the SEM results also showed that the cells on the hydrogel had a well-extended morphology (Figure 5D). VEGF and BMP-2 were coupled to the Gel–POSS hybrid hydrogels which were then implanted into rat calvarial defects, and the composite hydrogel showed a good drug-sustained release function (Figure 5C) while promoting angiogenesis and bone regeneration (Figure 5E).

### 5.4. Alginate

Alginate is a linear anionic biopolymer polysaccharide composed of homopolymeric units of 1,4-linked (-D-mannuronic acid) (M) and (-L-guluronic acid) (G) [66,67,108]. As a biopolymer from a natural source, alginate has many advantages, including biocompatibility, low toxicity, low immunogenicity and biodegradability [108,109], and thus can be a promising material for sustained-release drug delivery systems and tissue engineering. Alginate can form hydrogel through a complexation mechanism in the presence of cations (such as Ca^2+^, Mg^2+^, Mn^2+^, Co^2+^, Gu^2+^), and different ions have different effects on the gel-forming ability, rheological properties and gel structure of alginate [110]. Because of its relative safety and biocompatibility, Ca^2+^ is the widely used gelling agent of alginate hydrogel. Demineralized bone matrix (DBM) powder is a potential alternative bone graft material. Its composition and structure are similar to autologous bone. Because the powder contains a certain amount of bone morphogenetic protein, it can also produce osteoinductive effects, thereby promoting bone regeneration. Li [80] loaded demineralized bone matrix (DBM) powder and BMSCs into injectable self-healing alginate hydrogels, which is a cross-linked Schiff-base network as a self-healing component and a borax ion cross-linked physical network. With relatively strong mechanical properties and the loaded DBM powder and BMSCs, the hydrogels were found to promote effective bone regeneration and bone defect repair. Xie [111] prepared an icariin-laded hydroxyapatite/alginate porous composite scaffold (IC-HAA). The morphology and release behavior of the HAA and three IC/HAA scaffolds containing different concentrations of icariin were characterized by SEM and HPLC (Figure 6A,B). The results showed no significant differences in porosity or pore size between the four groups, and the release percentage increased with the increase of icariin loading. The results of the MTT assay showed that IC/HAA, IC and HAA could promote the proliferation of bone marrow mesenchymal stem cells (Figure 6C). ALP activity detection and quantification of mineralization showed IC-HAA scaffolds produced significant stimulation of the osteogenic differentiation of BMSCs (Figure 6D,E).

### 5.5. Hyaluronic Acid

Hyaluronic acid (HA) is a regular linear anionic polysaccharide belonging to a member of the glycosaminoglycan family. Its molecule is composed of two units (D-glucuronic acid and N-acetylglucosamine) connected by β(1,4) and β(1,3) glucosidic bonds repeatedly arranged. The relative molecular mass of HA ranges from 20,000 to millions of Daltons depending on the enzyme that catalyzes its synthesis [112]. HA is naturally widely present in many tissues and body fluids; the content varies greatly in different tissues and species [68,113]. As the most abundant component of synovial fluid in the joint cavity with viscoelastic, rheological and other physical and chemical properties, HA was initially used for the treatment of osteoarthritis [114]. In addition, HA also has special biological properties, such as inhibiting the growth of bacteria and interacting with extracellular matrix molecules and cell surface receptors (such as CD44, ICAM-1) to affect the tissue environment to regulate cell behavior [115,116]. Thus, hyaluronic acid has been widely used in various fields, such as ophthalmology, joint surgery, drug delivery and tissue engineering [117,118].

As an extracellular matrix molecule, HA tends to regulate and increase the regenerative properties of stem cells. When stem cells are encapsulated in a hyaluronic acid matrix, they will produce a synergistic effect to facilitate the tissue repair process [119]. Hydrogels constructed from HA are used more and more widely in various medical fields. HA is applied in the form of a hydrogel in bone tissue repair with good biocompatibility, low immunogenicity, degradability and excellent controlled-release properties. Zhang [81] prepared an injectable BMSC-laden hydrogel system which was formed by the enzyme-catalyzed crosslinking of hyaluronic acid-tyramine and chondroitin sulfate-tyramine (HA-CS hydrogel). Subsequently, the gelling rate, mechanical properties and degradation process of the scaffolds were characterized and optimized. The morphology of the hydrogels was observed by SEM, and the pores of 100–300 μm in the hydrogels could promote nutrient diffusion and provide sufficient space for cell proliferation (Figure 7A). In addition, the mechanical strength of hydrogels was investigated by monitoring the energy storage modulus (G′) and loss modulus (G″), while HA-CS hydrogel showed great compressive strength and fatigue resistance to cyclic compression (Figure 7B). In order to investigate the osteogenic differentiation of BMSCs within the HA-CS hydrogels, a quantitative analysis of the ALP was carried out. The result shows that the 3D scaffold could promote the osteogenic differentiation of BMSCs even under a complete culture medium due to their porous structure, and the group of BMSCs-laden hydrogel with BMP2 under osteoinductive medium could greatly improve the osteogenic differentiation efficiently (Figure 7C).

### 5.6. Fibrin

Fibrin is an insoluble protein formed from fibrinogen during the clotting of blood. It forms a fibrous mesh that impedes the flow of blood. As the natural scaffold formed by the action of thrombin on fibrinogen [120], fibrin can quickly initiate hemostasis and provide support for cell adhesion, migration, proliferation and cell-matrix useful for differentiation [69].

In the human body, the bone repair process begins with the formation of a blood clot, which can stop bleeding and serve as a temporary scaffold for the repair process. Inspired by the natural process, many researchers have taken blood clots as an ideal model for the design of biomimetic materials for bone tissue engineering [121]. The precursors of fibrin, fibrinogen and thrombin are mainly derived from the patient’s own blood, thus making fibrin less immunogenic, biocompatible and cheap. In addition, fibrin can be prepared in fibrin hydrogels with injectable properties and then solidified in situ; therefore, it can fill any irregularly shaped bone defects [122]. Further studies have also shown that fibrin alone cannot repair bone defects due to its fast degradation and weak mechanical properties. Some researchers have tried to use fibrin in combination with other materials, such as autografts [123], bioceramics and metals, to overcome these limitations. According to the current literature reports, composites of fibrin and bioceramics are commonly used. Nanohydroxyapatite [124], nano-calcium sulfate (nCS) [125], biphasic calcium phosphate ceramic (MBCP^®^) [126], wollastonite [127] and other materials have been reported to be used to form a composite with fibrin. The complex formed by bioceramics and fibrin is similar to autologous bone transplantation, which can promote the proliferation and differentiation of osteoblast-related cells, promote osteoinduction and improve the structure and function of newly formed bone. Liu [82] developed a pre-vascularized scaffold. They collected peripheral blood-derived mesenchymal stem cells (PBMSCs) and endothelial colony-forming cells (ECFCs) from peripheral blood. These cells were added to a fibrin gel and further mixed with polylactic acid-glycolic acid (PLGA) microspheres to form a fibrin gel/PLGA microsphere (FP) scaffold (Figure 8A). According to the three-dimensional reconstruction results from micro-CT, the CFP group led to superior bone regeneration compared with that observed in the FP group and the BC group, indicating that this type of pre-vascularized scaffold might promote bone healing at defect sites (Figure 8B). Although fibrin has unique advantages in the application of bone tissue engineering in theory, the use of fibrin is still controversial in actual applications due to large differences between product batches resulting from the different sources of fibrin and different separation techniques.

### 5.7. Silk Fibroin

Silk fibroin (SF) is a natural protein obtained by dewatering water-soluble sericin from the silk produced by silkworms [70,128]. Different from other natural biopolymers such as collagen, hyaluronate, fibrin and alginate, SF has stronger mechanical properties [129], a low degradation rate [130] and better biocompatibility. SF has provided a brand-new choice in the fields of controlled release, silk sutures and tissue engineering [70,130,131].

The silk fibroin aqueous solution can undergo a sol–gel transition when the pH is near the isoelectric point (PI) (3.8–3.9) of silk fibroin. Although SF can be gelled without adding a gelling agent, the low pH is often not conducive to drug loading. The insufficient cell adhesion ability and difficulty in cross-linking at a certain pH of SF can be improved through modifications and cross-linking on the amino acids with potentially active side groups. For example, RGD has been used to enhance the adhesion of SF to cells, and H_2_O_2_/horseradish peroxidase (HRP) to cross-link with tyrosine, which accounts for 10% of the total amino acid content of SF, to promote the cross-linking of SF. These modifications and cross-linking methods not only affect the delivery of drugs in SF by changing the interaction between drugs and SF, but also affect the interaction between SF and cells [132]. Cheng [133] incorporated the short silica nanoparticles (SiNPs)-distributed-silk fibroin nanofibers (SiNPs@NFs) into the SF hydrogel prepared above (SiNPs@NFs-SF). SF is used to simulate the ECM of natural bone, while SiNPs represent the mineral crystals in the extracellular matrix, and the addition of (SiNPs@NFs) increased the mechanical properties of the SF hydrogel (Figure 9A). The composite hydrogel provided a biocompatible micro-environment for cell adhesion, proliferation and osteogenic differentiation in vitro (Figure 9B,C). The formation of a large amount of new bone (Figure 9D) was confirmed by Micro CT on the large-sized cranial bone defect model, and the potential of the SiNPs@NFs-SF hydrogel for use in bone tissue engineering was primarily demonstrated. Yan [134] synthesized a small peptide NapFFRGD and used it to cross-link RGD-functionalized SF to make a hydrogel that can gel under physiological conditions (pH 7.4, 37 °C). The use of NapFFRGD peptide greatly reduced the gelation concentration, shortened the gelation time, improved the mechanical properties and increased the cell adhesion ability of SF. Despite the osteogenic potential of SF scaffold, more efforts are needed to improve osteoconductivity and promote the mineralization process. In recent years, SF is usually combined with an osteoconductive material such as Hap [135,136,137], nano-hydroxyapatite (nHA) [138,139,140,141], β-TCP [142], octacalcium phosphate (OCP) [143] and calcium phosphate cement (CPC) [144,145]. The composite scaffold can overcome the shortcomings of a single material with superior performance in bone defect repair.

### 5.8. Traditional Synthetic Biodegradable Polymers

Traditional synthetic materials, such as polyethylene oxide (PEO), polyethylene glycol (PEG) and polyvinyl alcohol (PVA), are usually easy to manufacture in industry, cheap and of high mechanical strength, but they are inferior to natural materials in terms of biocompatibility and some biological properties, which has limited their applications. Compared with natural materials, there has been less research on traditional synthetic materials on bone defect repair in recent years.

PEG is a synthetic polymer material formed by the ring-opening polymerization of ethylene oxide. PEG and its derivatives can form PEG gels under various gelation methods (physical, ionic, chemical or covalent interaction), and the PEG hydrogels have been widely used as a vehicle to control drug delivery and cell delivery to promote tissue regeneration [146]. In recent years, PEG, especially its derivatives, is still actively used due to the multi-functional tunability of macromolecular chemistry, strong mechanical properties, biodegradability, excellent biocompatibility and low price [71]. In bone tissue engineering, multi-arm star PEG polymers with modifiable end groups are prone to intermolecular cross-linking, forming a network structure and undergoing a solution–gel-like change, which are often used to prepare biocompatible hydrogels for regenerative medicine and tissue engineering [147]. For example, Nguyen [83] designed an eight-arm-PEG-acrylate (8-arm-PEG-A) hydrogel for local and continuous delivery of microRNA (miRNA) to block the expression of genes that have a negative impact on tissue regeneration and hMSCs being used to promote bone regeneration in rat skull defects. Liu [84] prepared polyethylene glycol diacrylate (PEGDA) hydrogel using the photoinitiation method, combined it with hydroxyapatite to make polyethylene glycol/hydroxyapatite (PEGDA/HA) mineralized hydrogel, and loaded it with Exendin4 (a kind of adjustable GLP-1 intestinal hormone analogue for bone growth and reconstruction). The effects of different calcium and phosphorus concentrations on the strength and Exendin4 release of PEGDA/HA hydrogels were investigated. They directly observed the micro-structure changes of PEGDA gel and PEGDA/HA gel by SEM (Figure 10A). PEGDA gel (Figure 10(Aa)) showed homogeneous and dense pore structure, and the pore size of PEGDA/HA gel (Figure 10(Ab)) was obviously larger than that of PNAGA gel. Compared with PEGDA/HA-3x gel (Figure 10(Ab)) and PEGDA/HA-5x gel (Figure 10(Ad)), the distribution of inorganic particles formed in the gel network of PEGDA/HA-4x (Figure 10(Ac)) was more homogeneous, and there were more inorganic particles. Therefore, PEGDA/HA-4x gel was chosen as the experimental group in the later experiments. In vitro release experiments showed that PEGDA gel and PEG-DA/HA-4x gel had similar release curves. Regardless of the initial concentration of Exendin4 solution, the release occurred within 1 day, reached the basic balance in about 5 days and continued for 20 days. PEGDA/HA-4x gel had a better sustained-release effect than PEGDA gel (Figure 10B). The hydrogel was implanted into a rat bone defect model, and histological observation showed that the hydrogels had good biocompatibility and could promote the formation of bone (Figure 10C,D).

### 5.9. Ion-Complementary Self-Assembling Peptide

Zhang [148] reported the first self-assembling peptide (SAP) EAK16 in a yeast protein *Zuotin* in 1993 in researching Z-DNA-binding proteins. With the in-depth research of scientists, there has been a significant advance in the field of self-assembling peptides, which can form assemblies by hydrogen bond formation, hydrophobic and electrostatic interactions, van der Waals forces and π-π stacking. RADA16-I, designed by Yokoi, is the most widely researched self-assembling peptide.

Ion-complementary self-assembling peptides represented by RADA16-I have special amphiphilic characteristics due to their alternate arrangement of hydrophobic amino acids/hydrophilic amino acids in their sequences [149]. In an aqueous solution, such short peptides can form a stable β-sheet structure. Under physiological conditions, the aqueous solution of such short peptides can quickly form a hydrogel with pores from 5 to 200 nm in size and a water content of more than 99%. Thus, the ion-complementary self-assembling peptide materials have shown considerable development potential in cell culture, tissue engineering and drug delivery systems [72,73]. As an extracellular matrix bio-scaffold material, RADA16 hydrogel has been used to repair cartilage defects and brain damage and promote bone regeneration [150], which can significantly promote the adhesion, proliferation and osteogenic differentiation of seed cells such as MSCs [151], MC3T3 [152] and other seed cells. Materials with higher biological activity and higher mechanical strength can be designed by covalently binding some functional groups, such as biotin, protein sequences and extracellular matrix mimic sequences, to the C-terminus or N-terminus of RADA16. He [85] designed hydrogels loaded with basic fibroblast growth factor (bFGF) using d-RADA16 and l-RADA16. Studies indicated that d-RADA16 hydrogels show certain potential in loading and releasing bFGF, which can further promote bone regeneration based on micro-CT analysis. RADA16 is a self-assembling peptide material with good bioactivity. To improve the bioactivity of a material, some specific functional motifs can be added to its peptide sequence. Tian [78] prepared a kind of RADA16-RGD hydrogel which can carry and control the release of two growth factors, VEGF and BMP-2, at the same time. As shown in the SEM images (Figure 11A), both RADA16 and RADA16-RGD could self-assemble to form 3D network structures, and the pore sizes of these nanofibers ranged from 30 to 200 nm for RADA16 and 40–250 nm for RADA16-RGD. The CCK-8 test results showed that these materials have good biocompatibility (Figure 11C). Further relevant experimental results showed that RADA16-RGD showed a better performance than RADA16 in promoting cell proliferation, adhesion and bone formation. In addition, RADA16-RGD can effectively control the release of VEGF and BMP-2 (Figure 11B). Li [153] designed the biological composite scaffold RADA-CPC composed of RADA and calcium phosphate cement (CPC). The composite scaffold has suitable pore size and good biocompatibility, which can simulate an extracellular matrix, and therefore has good cell adhesion and good osteogenic ability. In addition, more active alkaline phosphatase (ALP) and more mineralized crystals were detected in this scaffold. This new type of RADA/CPC composite scaffold has potential application prospects in the treatment of bone defects.

## 6. Conclusions

Much research has been carried out on hydrogel materials as sustained-release drug carriers for bone defect repair, but the final clinical effects are still far from satisfactory, and the treatment of large bone defects is still a major challenge. Natural biodegradable materials such as collagen, chitosan, sodium alginate and gelatin still have the disadvantages of weak mechanical strength, poor osteoconductivity, immune rejection and uncontrollable degradation. Synthetic degradable hydrogel-forming polymers have disadvantages such as slow degradation rate, high immunogenicity and poor biocompatibility. Natural tissues are structurally heterogeneous, their mechanical properties are often nonlinear and they can usually self-heal in response to injury. In contrast, many hydrogel materials, especially those formed from synthetic polymers, are typically linearly elastic in their mechanical properties and therefore do not perfectly mimic the structural and mechanical properties of natural tissues. Thus, both natural and synthetic hydrogels as sustained-release drug carriers for bone defect repair need to be further studied to address their disadvantages.

As a new type of biological material, self-assembling peptide hydrogels can provide a suitable microenvironment for cell proliferation and differentiation due to their suitable pore size. The degradation rate of the self-assembling hydrogels in the body may match the growth rate of bone tissue, which is conducive to the formation of new bone tissue, as well as good osteoconductivity, osteoinductivity and osteogenic properties. In addition, the potential of the self-assembling hydrogels as slow-release carriers for small molecule drugs and cytokines has been demonstrated by many researchers. Thus, self-assembling hydrogels can be promising sustained-release drug carriers in bone defect repair. Intensive in-depth research is needed to optimize the peptide sequence of the self-assembling peptides or combine other polymer materials to form composite materials to obtain stronger mechanical properties to improve mechanical performance of the self-assembling peptide hydrogels, making them more widely and deeply applied to bone defect repair.

## Figures and Tables

**Figure 1 polymers-14-04906-f001:**
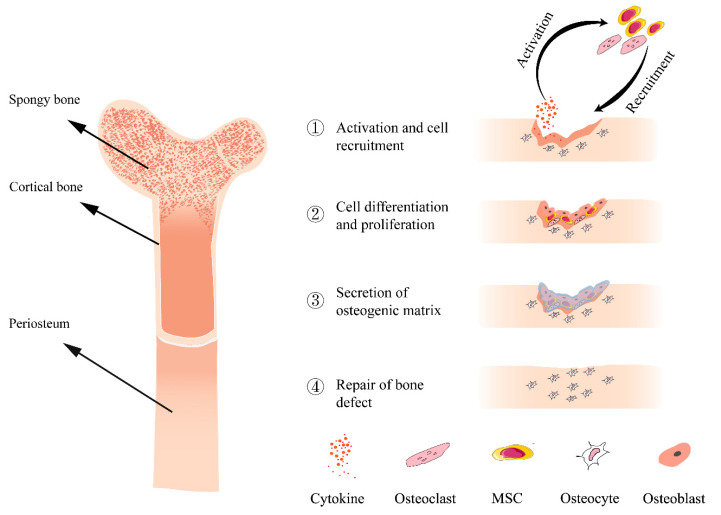
Schematic diagram of the structure of bone (**left**) and bone repair process (**right**).

**Figure 2 polymers-14-04906-f002:**
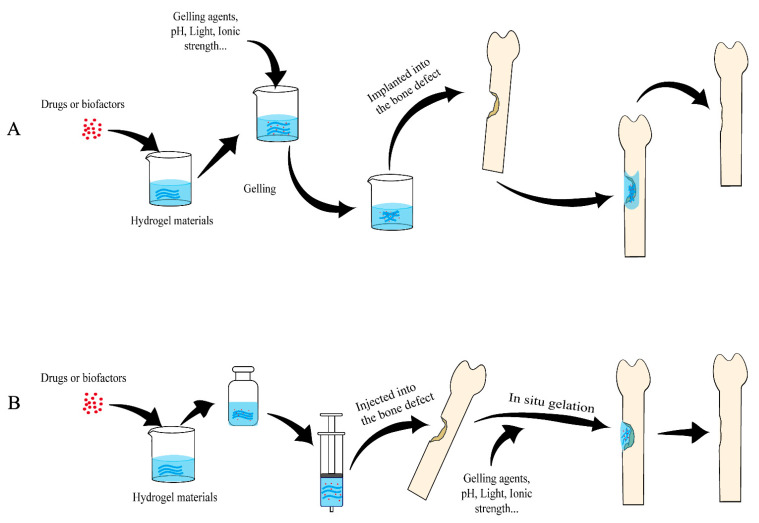
Schematic diagram of applying hydrogels as sustained-release drug carriers to repair bone defects. (**A**) Hydrogels formed in vitro being implanted into the bone defects. (**B**) Hydrogels formed in situ. Hydrogel materials are mixed with drugs and/or active biofactors (may also include cells) to form hydrogels and slowly release the drugs or biofactors locally to repair bone defects.

**Figure 3 polymers-14-04906-f003:**
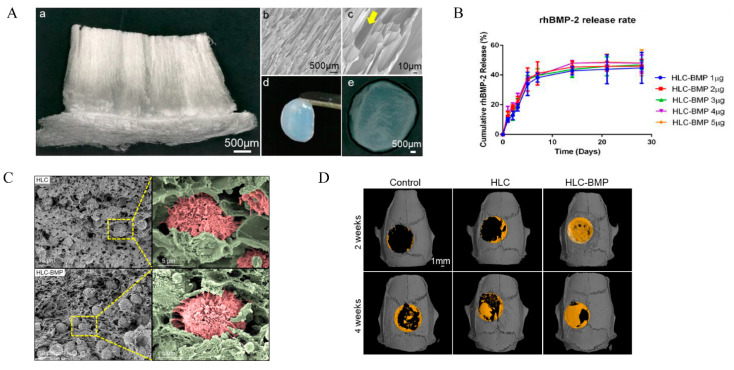
The HLC/HLC-BMP sponges for bone repair. (**A**) Gross view or SEM micrographs of the HLC/HLC-BMP sponges; (**a**) Gross view of the lyophilized HLC sponge, using a digital microscope at 30× magnification; (**b**,**c**) SEM micrographs of lyophilized HLC sponge at 100× and 500× magnification; Yellow arrow indicates the unique vertical pore structure of the HLC sponge; (**d**,**e**) Gross view of the HLC-BMP hydrogel. (**B**) Cumulative rhBMP-2 release kinetics from the lyophilized HLC-BMP sponges. (**C**) SEM images of MSCs attached in the HLC and HLC-BMP sponges after culturing for 4 h. (**D**) Micro-CT of the rat cranial defect repair after 2 and 4 weeks of implantation of hydrogels. Adapted with permission from Ref. [76]. Copyright 2022 Biomolecules.

**Figure 4 polymers-14-04906-f004:**
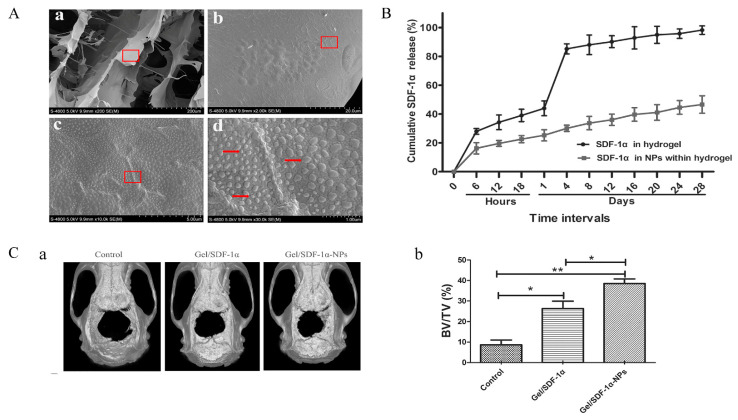
The SDF-1α/CS/CMCS NPs incorporated CS/GP hydrogels for bone repair. (**A**) SEM observation of SDF-1α/CS/CMCS NPs incorporated CS/GP hydrogels. (**B**) In vitro release of SDF-1α from CS/GP hydrogels and SDF-1α from SDF-1α/CS/CMCS NPs incorporated CS/GP hydrogels. (**C**) The micro-CT evaluation of in vivo bone formation. (**a**) Representative 3D reconstruction images of different groups. (**b**) Bone volume/total volume (BV/TV) varied in different groups (* *p* < 0.05, ** *p* < 0.01). Adapted with permission from Ref. [79]. Copyright 2022 International Journal of Bio-logical Macromolecules.

**Figure 5 polymers-14-04906-f005:**
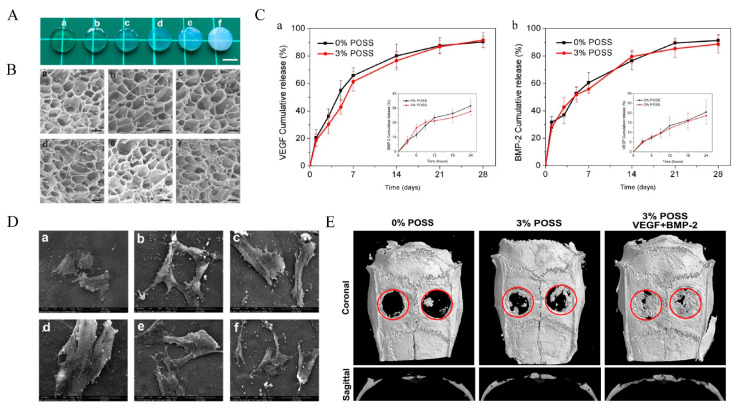
The Gel–POSS hybrid hydrogels for bone repair. (**A**) Optical appearance and (**B**) microstructure after freeze-drying as observed by the SEM of (**a**) pure Gel hydrogel and (**b**–**f**) 1–5% POSS. (**C**) (**a**) Sustained release curve of VEGF and (**b**) BMP-2 at early stage (within 24 h) and late stage. (**D**) SEM scanning of the cells on the hydrogels on day 3. (**E**) Representative coronal and sagittal Micro-CT images of calvarial bone defects after 8 weeks of implantation. Adapted with permission from Ref. [77]. Copyright 2022 American Chemical Society.

**Figure 6 polymers-14-04906-f006:**
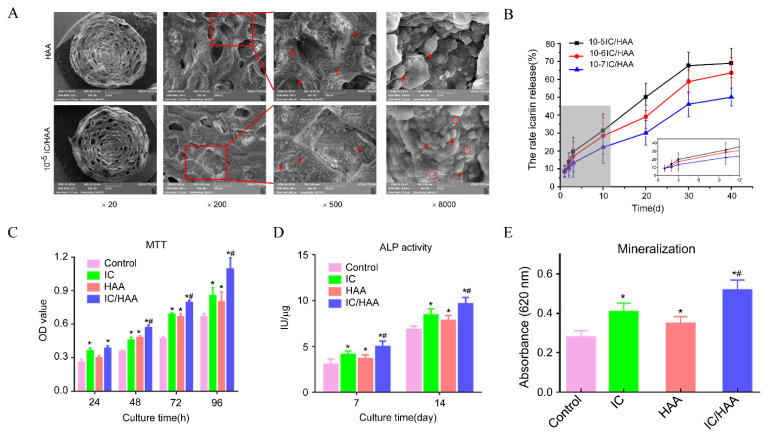
The IC/HAA scaffold for bone repair. (**A**) The SEM images of the IC/HAA scaffolds. Red arrow indicates the nanohydroxyapatite. Red circle indicates the icariin. (**B**) HPLC detection of icariin release from IC/HAA composite scaffolds. (**C**) The optical density (OD) values of the MTT assays in the four groups. (**D**) ALP activity in each group. (**E**) Mineralization detected by measurement of the absorbance at 620 nm. Notes: * *p* < 0.05, compared with the control group, # *p* < 0.05, compared with the HAA group. Adapted with permission from Ref. [111]. Copyright 2022 International Journal of Nanomedicine.

**Figure 7 polymers-14-04906-f007:**
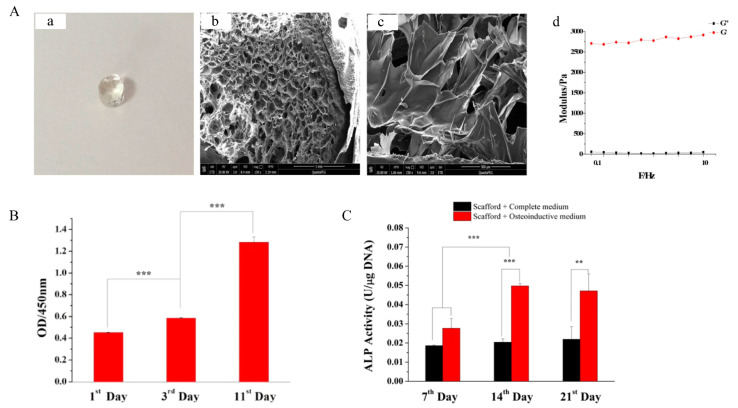
The HA-CS hydrogels for bone repair. (**A**) The morphology of HA-CS hydrogel (**a**–**c**), the storage modulus (G′) and loss modulus (G″) of hydrogel (**d**). (**B**) The proliferation of BMSCs within hydrogels at certain time intervals. (**C**) The ALP activity of BMSCs osteogenic differentiation within HA-CS hydrogels in different groups. Adapted with permission from Ref. [81]. Copyright 2022 Materials Science & Engineering C. ** *p* < 0.01, *** *p* < 0.001.

**Figure 8 polymers-14-04906-f008:**
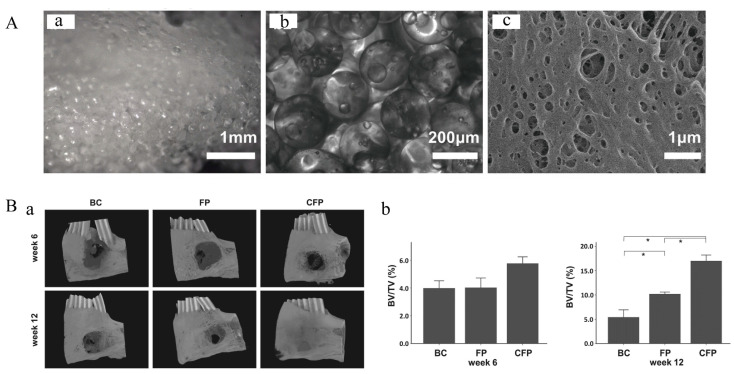
The FP scaffold for bone repair. (**A**) Morphology and structure of FP scaffold. (**a**) Stereomicroscopy image of FP scaffold. (**b**) Image of FP scaffold under an inverted microscope. (**c**) SEM image of FP scaffold. (**B**) Micro-CT evaluation of in vivo bone growth within harvested mandible samples. (**a**) Three-dimensional reconstructed images at week 6 and 12. (BC: Blank control, bone defect without implant; FP: Bone defect implanted with FP scaffold; CFP: Bone defect implanted with cell-laden FP scaffolds). (**b**) Results of quantitative analyses of samples harvested at weeks 6 and 12. The symbol * denotes a statistical significance between two groups (*p* < 0.05). Adapted with permission from Ref. [82]. Copyright 2022 NPG Asia Materials.

**Figure 9 polymers-14-04906-f009:**
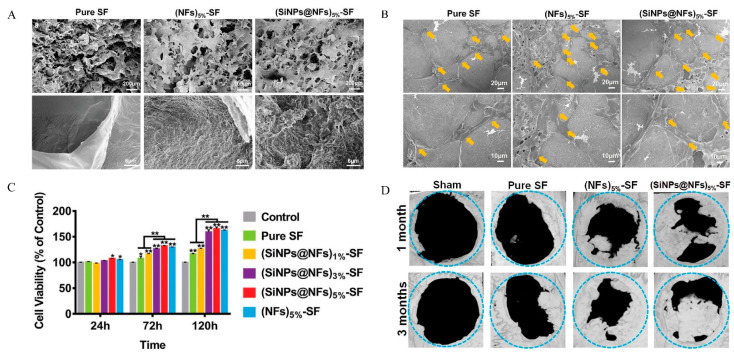
The SiNPs@NFs-SF hydrogels for bone repair. (**A**) SEM images of the Pure SF, (SiNPs@NF)5%-SF and (NFs) 5%-SF hydrogels. (**B**) The SEM images of adhesion and spreading of the MC3T3-E1 cells cocultured with the composite hydrogels (yellow arrows represent the cell nucleus). (**C**) The viability and proliferation of the MC3T3-E1 cells cocultured with the composite hydrogels, *n* = 5, * *p* < 0.05, ** *p* < 0.01. (**D**) Micro-CT images of rat cranial defects after implantation of the composite hydrogels for 1 and 3 months. Adapted with permission from Ref. [133]. Copy-right 2022 Advanced Healthcare Materials.

**Figure 10 polymers-14-04906-f010:**
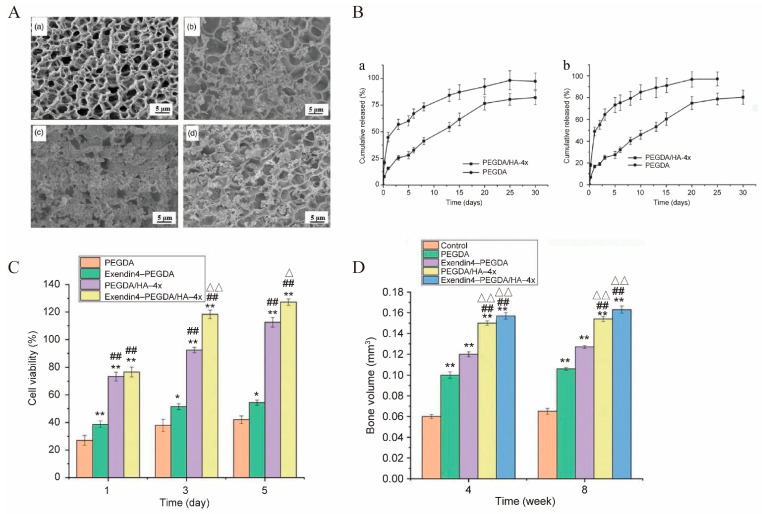
The PEGDA gel for bone repair. (**A**) SEM of PEGDA gel (**a**), PEGDA/HA-3x gel (**b**), PEGDA/HA-4x gel (**c**) and PEGDA/HA-5x gel (**d**). (**B**) Exendin4 cumulative release curve of PEGDA gel and PEGDA/HA-4x gel within 30 d; initial Exendin4 concentration was 5 μg/mL (**a**) and 10 μg/mL (**b**). (**C**) Proliferation of L929 cells grown on gel surface on 1, 3 and 5 d, and (**D**) new bone volume. (compared with the PEGDA, * *p* < 0.05 and ** *p* < 0.01; compared with Exendin4-PEGDA, ## *p* < 0.01; compared PEGDA/HA-4x group, Δ *p* < 0.05 and ΔΔ *p* < 0.01). Adapted with permission from Ref. [84]. Copyright 2022 Journal of Biomaterials Applications.

**Figure 11 polymers-14-04906-f011:**
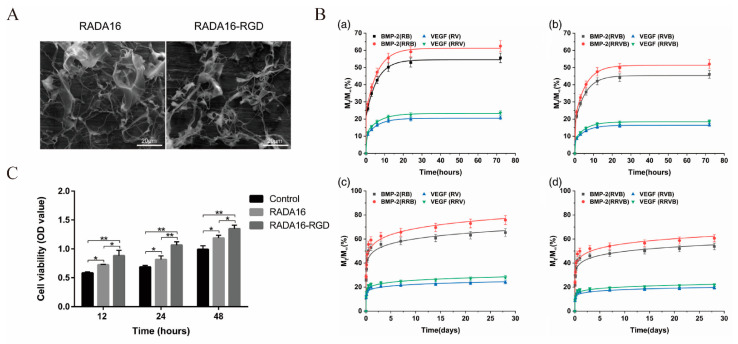
The RADA16 and RADA16-RGD hydrogels for bone repair. (**A**) The SEM images of RADA16 and RADA16-RGD. (**B**) Release profiles of growth factors in vitro. VEGF and BMP-2 released from (**a**) single-factor carriers and (**b**) dual-factor carriers during the first 72 h. VEGF and BMP-2 released from (**c**) single-factor carriers and (**d**) dual-factor carriers during the 28-day period. (**C**) Proliferation of cells on different samples evaluated by CCK-8 assay (*n* = 6, * *p* < 0.05, ** *p* < 0.01). Adapted with permission from Ref. [78]. Copyright 2022 Journal of Biomedical Materials Research Part A.

**Table 1 polymers-14-04906-t001:** The hydrogel materials involved in this paper.

Hydrogels	Source	Biocompatibility	Degradation	Influencing Factors of Mechanical Properties	Activity	Ref.
Collagen	Natural	Good	Rapid	The rate of degradation	Osteogenesis	[20,62,63]
Chitosan	Natural	Good	Depending on the molecular weight and the acetylation degree	Crosslinking methods	Antibacterial activity, Osteoinduction	[64]
Gelatin	Natural	Good	Rapid	Crosslinking methods	Cell adhesion	[65]
Alginate	Natural	Good	Slow and uncontrollable	Ion type, sequence and composition of the alginate chain	Osteoinduction	[66,67]
Hyaluronic acid	Natural	Good	Rapid	Size and concentration	Osteoinduction	[68]
Fibrin	Natural	Good	Rapid	The rate and duration of loading	Osteogenesis	[69]
Silk fibroin	Natural	Good	Slow	Processing techniques, com-position, matrix stiffness, β-sheet content, scaffold morphology and topology	Cell adhesion	[70]
PEG	Synthetic	Good	Depending on molecular weight	Molecular weight	Cell adhesion	[71]
RADA16	Synthetic	Good	Rapid	Concentration	Osteogenesis	[72,73]

**Table 2 polymers-14-04906-t002:** The drugs, biofactors and cells in hydrogel carriers involved in this paper.

Drugs/Biofactors/Cells	Category	Effects Related to Bone Repair	Carriers	Method of Ad-Ministration	Ref.
Alendronate	Third-generation bisphosphonate	Inhibiting the activity of osteoclasts	Collagen	Implant	[74,75]
Bone morphogenetic protein-2 (BMP-2)	Transforming growth factor-β	Osteoinduction	Collagen, Gelatin, RADA	Implant	[76,77,78]
Stromal cell-derived factor-1α (SDF-1 α)	Chemokine	Stimulating the recruitment of stem cells	Chitosan	Inject	[79]
Vascular endothelial growth factor (VEGF)	Growth factor	Promoting angiogenesis	Gelatin, RADA	Implant, Inject	[77,78]
Bone marrow stromal stem cells (BMSCs)	Stem cell	Osteogenic differentiation	Alginate, Hyaluronic acid, Fibrin	Inject	[80,81,82]
MicroRNA (miRNA)	Nucleic acid	Blocking gene expression that negatively affects tissue regeneration	PEG	Implant	[83]
Exendin 4	GLP-1 intestinal hormone analogue	Regulation of bone growth and reconstruction	PEG	Implant	[84]
Basic-fibroblast growth factor (bFGF)	angiogenic factor	Promoting cell proliferation and migration	RADA	Inject	[85]

## Data Availability

Not applicable.

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
