# Peer review of "Application of Hydrogels as Sustained-Release Drug Carriers in Bone Defect Repair"

_polymers, 2022, doi:10.3390/polym14224906_

Round 1

Reviewer 1 Report

The present review paper is well written and organized. The general structure and flow is clear for the readers. I suggest to the authors to add some graphical image for the second paragraph (Bone defect repair) to simplify the readers. 

Moreover I suggest to add a paragraph on different materials used as bone substitutes (for e.g. doi: 10.3390/ijms22105236;  10.3390/ma12132157).

Reviewer 2 Report

I have the following comments/suggestions:

1. Include a table listing the mechanical properties of the native tissue and how each of the hydrogen scaffolds compare in terms of mechanical behavior.

2. Include images of cell proliferation and SEM micrographs of different polymer scaffolds to indicate what optimum conditions support bone regeneration.

3. Include figures for as many sections as possible 

4. Do a complete grammar check/proof reading and ensure that there are no grammatical errors or redundancies in the written content.

Reviewer 3 Report

The authors presented an overview of hydrogels in bone repair. Starting from an introduction to the bone regeneration process and the current treatment options, the authors then introduce hydrogels of different materials composition, their pros and cons, and their potential as drug carriers. A few comments as below:

Table 1 and 2: please include citations for each hydrogel presented.

Table 1 and 2: which biofactor has been loaded into which hydrogel? it would be nice to mention it in Table 2.

5.1 Collagen: collagen has very different mechanical properties depending on its assembly of it into fibrils or fibers, and the loading direction. So it is not necessarily a weak material. Please clarify in which state the collagen is and under which loading condition.

5.3 Gelatin: second paragraph, (1) how does GO crosslink gelatin? how does it improve its mechanical properties? it is not stated in the text. The text is more about biocompatibility and regenerative function. (3) gelatin-chitosan is not a hybrid. hybrid usually refers to a composite with more than one category of materials (metal, ceramic, polymer). Bioceramic/gelatin is a hybrid.

5.1-5.3: it would make more sense to judge/compare the mechanical properties of these biopolymer hydrogels when the chain length, percentage of them in the hydrogel (some could dissolve much more than the other), and crosslinking/gelling condition. And these need to be supported with data from certain references. It also depends on the way to apply these hydrogels: by implantation the hydrogel doesn't have to show proper rheological properties, thus can be much stiffer; alternatively, if the hydrogel is applied through injection, it is difficult to balance the mechanical properties and the injectability.

5.1-5.8 it would be nice to know how each gel is incorporated into existing treatment procedures, or whether they are an independent, stand-alone procedure. If possible, please discuss this in combination with Figure 1.

One important aspect of drug carrier, except the possibility of carrying different factors, is the control of the release through the properties of the hydrogel. It would be nice to provide a general idea of the approaches to tune the release rate of the drugs/factors in a hydrogel.  

Round 2

Reviewer 2 Report

Thanks for addressing all of my suggestions/comments. Great job!

Author Response

Thanks for your  kind comments and encouragement on our revision.

Reviewer 3 Report

The authors had properly addressed all comments. 

Author Response

Thanks for your kind comments and suggestions on our revision.